# Fine-Mapping of the Human Blood Plasma N-Glycome onto Its Proteome

**DOI:** 10.3390/metabo9070122

**Published:** 2019-06-26

**Authors:** Karsten Suhre, Irena Trbojević-Akmačić, Ivo Ugrina, Dennis O. Mook-Kanamori, Tim Spector, Johannes Graumann, Gordan Lauc, Mario Falchi

**Affiliations:** 1Department of Physiology and Biophysics, Weill Cornell Medicine-Qatar, Education City, PO 24144, Doha, Qatar; 2BICRO BIOCentar, Glycoscience Research Laboratory, Genos Ltd., Borongajska cesta 83H, 10000 Zagreb, Croatia; 3Department of Clinical Epidemiology, Leiden University Medical Centre, P.O. Box 9600, 2300 RC Leiden, The Netherlands; 4Department of Twin Research and Genetic Epidemiology, King’s College London, London SE1 7EH, UK; 5Scientific Service Group Biomolecular Mass Spectrometry, Max Planck Institute for Heart and Lung Research, W.G. Kerckhoff Institute, Ludwigstr. 43, D-61231 Bad Nauheim, Germany

**Keywords:** glycomics, proteomics, N-glycosylation, population study, aptamers, HILIC-UPLC, SOMAscan

## Abstract

Most human proteins are glycosylated. Attachment of complex oligosaccharides to the polypeptide part of these proteins is an integral part of their structure and function and plays a central role in many complex disorders. One approach towards deciphering this human glycan code is to study natural variation in experimentally well characterized samples and cohorts. High-throughput capable large-scale methods that allow for the comprehensive determination of blood circulating proteins and their glycans have been recently developed, but so far, no study has investigated the link between both traits. Here we map for the first time the blood plasma proteome to its matching N-glycome by correlating the levels of 1116 blood circulating proteins with 113 N-glycan traits, determined in 344 samples from individuals of Arab, South-Asian, and Filipino descent, and then replicate our findings in 46 subjects of European ancestry. We report protein-specific N-glycosylation patterns, including a correlation of core fucosylated structures with immunoglobulin G (IgG) levels, and of trisialylated, trigalactosylated, and triantennary structures with heparin cofactor 2 (SERPIND2). Our study reveals a detailed picture of protein N-glycosylation and suggests new avenues for the investigation of its role and function in the associated complex disorders.

## 1. Introduction

Protein glycosylation is a ubiquitous form of protein modification and plays a central role in many biological processes [1]. Glycans have been proposed to be directly involved in the pathophysiology of many major diseases [2]. Although carbohydrates are one of the major biological building blocks, understanding of the human glycan code lags far behind that of DNA, RNA, and proteins. High-throughput characterization of blood samples from large population studies using deep molecular phenotyping technologies can provide an unbiased view of the participant’s health state and associated variations in their molecular blood composition may thereby reveal novel molecular biomarkers for diagnosis and disease progression. The resulting multi-omics data sets allow for the identification of functional relationships between molecular traits and their mapping onto potentially health-relevant pathways [3], as exemplified by recent studies on the human blood metabolome–transcriptome interface [4], the link between the blood methylome and metabolome [5], and genome-wide association studies with the blood proteome [6] and metabolome [7]. Here we extend this approach to investigate the relationship between natural variation observed in the blood circulating proteome and its associated N-glycome. Both types of deep molecular phenotypes (“-omics”) have only recently become accessible to high-throughput measurement. To the best of our knowledge this is the first study of the relationship between the blood proteome and its N-glycome and may shed new light on the specific glycan garments of blood circulating proteins.

Due to their high concentration range, blood circulating proteins are intrinsically difficult to assess by mass-spectrometry. We therefore deployed a highly multiplexed, sensitive, quantitative, and reproducible proteomics tool (SOMAscan™, Boulder, CO, USA) [8]. The SOMAscan assay is based on a new generation of protein-capture aptamers (slow off-rate modified aptamer, SOMAmer™, City, Country). These SOMAmers include naturally occurring and chemically modified nucleotides and are selected from large randomized nucleic acid libraries to bind specifically to proteins implicated in numerous diseases and physiological processes and to target a broad range of secreted, intracellular, and extracellular proteins. The assay used in this study determines the levels of 1129 native proteins in complex matrices by transforming each individual protein concentration into a corresponding SOMAmer concentration, which is then quantified by microarray. It offers a high dynamic range and allows quantifying proteins over eight orders of magnitude in abundance (from femtomolar to micromolar), with low limits of detection (38 fM median LOD) and excellent reproducibility (5.1% median % CV). To date, the SOMAscan assay has been applied successfully to biomarker discovery and validation in many pharmaceutical research and development projects, diagnostics discovery and development projects, and academic research projects, including Alzheimer’s disease [9,10], Duchenne muscular dystrophy [11], aging [12], cancer [13,14], cardiovascular disease [15], genome-wide association studies [6], and in combination with mRNA sequencing [16]. 

N-glycosylation is a co- and post-translational protein modification process that influences the structure, cellular localization, and function of the majority of human proteins [17,18]. It consists in the covalent linkage of complex oligosaccharides to specific protein sites, and many proteins only become functional after glycosylation. Variation in the glycome make-up associates with disease development [1] and response to medication [19]. Moreover, glycans have been shown to constitute markers of chronological and biological age [20], Parkinson’s disease [21], and kidney function [22]. While the protein sequence is unambiguously coded by the respective gene, the glycan configuration is flexible and influenced by the environment [23]. Analysis of the glycoconjugates is challenging due to the complex branched structures of the sugars and their comparative fragility. Thanks to recent advances in high-throughput technologies the systematic analysis of the glycome composition in large population cohorts and clinical studies is now feasible [24]. For the study of total plasma or serum N-glycomes, glycans are released from total plasma proteins using denaturation and enzymatic cleavage, followed by labelling with 2-aminobenzamide and profiling using hydrophilic interaction ultra-performance liquid chromatography (HILIC-UPLC). 

In this study we analysed the correlations between proteomics and N-glycomics measurements made in blood samples of participants of the QMDiab and the TwinsUK cohort (see methods) under the assumption that a correlation between a specific protein and N-glycan is indicative of one of the following: a direct physical link between both, e.g., cases where the protein is a major source of the observed N-glycan, the protein controls the production of the N-glycan, e.g., an enzyme that acts in the biosynthesis of a specific N-glycan, the N-glycan controls the abundance of the protein in the blood, e.g., through regulation of translocation processes, or both protein and N-glycan levels are controlled by a confounding factor, like disease state, smoking, or age. As this is the first study of its kind, we focus on protein–glycan associations that are replicable and robustly observed in different ethnicities.

## 2. Results

### 2.1. Correlation between Plasma N-Glycans and Blood Circulating Protein Levels 

Using the SOMAscan^TM^ platform we quantified the levels of 1129 proteins in blood EDTA plasma samples of 352 participants of the Qatar Metabolomics Study on Diabetes (QMDiab) [25] and in parallel determined the total plasma N-glycome for 344 of these samples (a smaller sample numbers of N-glycan measurements was due to limited sample availability). For replication, we included data from 46 participants of the TwinsUK study for whom both SOMAscan and N-glycan data was available. In total, 1116 protein and 113 glycan traits overlapped between both studies and were further analysed (Appendix A). We used inverse-normal scaled glycan and protein levels and linear models to determine the correlation between both phenotypes in the QMDiab cohort and found 834 protein–glycan pairs that associated with each other at a conservative Bonferroni level of significance (*p* < 3.96 × 10^−7^ = 0.05/1116/113). We attempted replication in the TwinsUK cohort; 145 out of the 834 Bonferroni significant associations from the discovery study were nominally significant (*p* < 0.05) and displayed a consistent trend in the TwinsUK study. These associations covered 62 glycan traits and 43 proteins. Only two associations with a nominal *p*-value (*p* < 0.05) in TwinsUK showed a conflicting trend (Appendix A). We further arranged the replicated 145 associations into a matrix of 62 glycan by 43 protein with the respective correlation r^2^ values and direction of association as entries (Figure 1, Appendix A). Filtering of this matrix by protein or glycan trait(s) revealed specific protein glycosylation patterns of which we summarize the most salient features next (Table 1 and Figure 2). 

### 2.2. Immunoglobulin G

The strongest positive associations between immunoglobulin G (IgG, aptamer identifier: SL000467) were with the percentage of core fucosylated N-glycan structures (FUC-C, glycan identifier: PGP93) and with N-glycan structures containing bisecting GlcNAc in total plasma N-glycans (Btotal, PGP108), and with the percentage of fucosylation of digalactosylated N-glycan structures in total neutral plasma N-glycans (FG2n total /G2n, PGP75). This observation is in agreement with the well-established fact that one of the most abundant IgG N-glycoforms is the fucosylated digalactosylated form and that more than 90% of IgG N-glycans contain core fucose [26,27]. IgG N-glycans are predominantly complex biantennary glycans and although glycoforms with bisecting GlcNAc make 5–10% of total IgG N-glycans pool, it is known that bisecting suppresses further branching. Bisecting GlcNAc is therefore limited to complex N-glycans with up to two antennas. The observed positive protein–glycan associations in this case hence reflect the dominant N-glycan garments of the IgG proteins and the fact that IgG is the major contributor of core fucosylated structures to the plasma N-glycome. 

We also observed strong negative associations with IgG, i.e., with the percentage of mannose-rich M9 (GP18n, PGP69) and with biantennary digalactosylated A2G2 (GP8n, PGP65) in neutral plasma glycans, as well as with sialylated forms of biantennary digalactosylated A2G2S(6)1 + A2G2S(3)1 (PGP14) N-glycans in total plasma N-glycans. As N-glycans from IgG likely dominate the N-glycan pool, these negative associations suggest that the relative contributions of other dominant N-glycans decrease. For instance, Apo B-100 is probably the protein that most highly contributed with M9 to the total plasma N-glycans pool [28,29]. We did not observe any complementary positive associations for M9, likely because the relevant proteins were not targeted by the SOMAscan assay.

### 2.3. Immunoglobulin M

The strongest association of immunoglobulin M (IgM, SL000468) was with the percentage of FA2BG2S(3)1 + FA2BG2S(6)1 in total plasma glycans (PGP16) and the percentage of FA2BG2 in total plasma glycans (PGP11), followed by the percentage of mono-sialylation of core-fucosylated digalactosylated structures without bisecting GlcNAc in total plasma N-glycans (PGP42). FA2BG2S1 glycans (S(3) and S(6) variants could not be separated under the chromatographic conditions used) are dominant N-glycan structures on IgM and confirm previous knowledge. The positive association of FA2BG2 with IgM was somewhat unexpected since it represents only around 4% of the IgM glycan pool [30]. We also observed a positive association of IgM with N-glycan structures containing bisecting GlcNAc in total plasma N-glycans (Btotal, PGP108), paralleling that of these glycans with IgG and also with CD5 antigen-like (CD5L). The strongest association of CD5L was also with FA2BG2S1 glycans, which is consistent with the observation that an IgM-associated peptide, later identified as CD5L, was found in all IgM fractions purified from plasma or serum by various methods [31].

### 2.4. Heparin Cofactor 2

The strongest associations for heparin cofactor 2 (SERPIND2, SL004466) were with the percentage of trisialylated (PGP97), trigalactosylated (PGP102), and triantennary (PGP105) structures in total plasma N-glycome, and specifically with the percentage of A3G3S(3,6)2 in total plasma N-glycans (PGP22). Anticorrelations were with the percentage of agalactosylated structures in total plasma N-glycans (PGP99) and with the percentage of antennary-fucosylation of tetragalactosylated structures in total plasma N-glycans (PGP113). Not much is known about human SERPIND2 glycosylation, but observations of diantennary and triantennary glycans on SERPIND2 in hamster ovary cells support our findings [32].

### 2.5. Alpha-(1,3)-Fucosyltransferase 5

Levels of alpha-(1,3)-fucosyltransferase 5 (FUT5; SL014008) associated with the percentages of A3G3S(3,3,3)3 (PGP24), A4G4S(3,3,3)3 (PGP30), A4F1G3S(3,3,3)3 + A4F1G3S(3,3,6)3 + A4F1G3S(3,6,6)3 (PGP32), and A4F1G4S(3,3,3,6)4 (PGP36) and consequently also with derived N-glycan traits, i.e., tetraanntenary structures (PGP106), the ratio of trisialylated and tetrasialylated tetragalactosylated structures (PGP110), tetragalactosylated (PGP103), antennary fucosylated structures (Fuc-A, PGP92), and antennary-fucosylation of trigalactosylated structures (PGP112). GP32 and GP36 contain antennary fucose moieties and are hence expected to correlate with FUT5 activity. On the other hand, positive associations of triantennary and tetraantennary sialylated N-glycans without antennary fucose are more challenging to explain, since as potential substrates of FUT5 one would expect them to be negatively correlated. However, much of the regulation of N-glycosylation remains enigmatic However, we still do not know much about N-glycosylation regulation. It is possible that if the amount of substrate increases, FUT5 abundance is upregulated as a response, but fucosylation activity itself is inhibited (for example sterically). Furthermore, FUT5 itself is a glycoprotein but to the best of our knowledge no detailed information about the type of glycans it carries is available. Interestingly, a recent GWAS with N-glycans, identified SNP rs1169303 near HNF1A as associated with PGP30 [33]. HNF1A has been shown to co-regulate the expression of most fucosyltransferases, including FUT5 [34]. Together, the glycan associations with FUT5 hence likely reflect FUT5 activity.

### 2.6. C-Reactive Protein

The glycan associations involving C-reactive protein (CRP; SL000051) are driven by the percentage of trisialylated structures (PGP97), and in particular by the specific trisialylated N-glycan A3F1G3S(3,3,3)3 + A3F1G3S(3,3,6)3 (GP29). CRP is an important acute-phase protein and associated with the future occurrence of coronary events [35]. As we have joint measurements of N-glycans and CRP using clinical biochemistry methods available for 798 individuals in the TwinsUK study, we could replicate these associations in a larger population sample (Appendix A). In this case we used QMDiab for replication. We found excellent agreement between both studies: of the 23 strongest CRP-glycan associations observed in TwinsUK, only one was not significant (*p* < 3.96 × 10^−7^) in QMDiab, while only four out of the 90 remaining weaker association were replicated in QMDiab. All replicated associations had consistent effect sizes (Figure 3).

### 2.7. Other Proteins

Low affinity immunoglobulin gamma Fc region receptor III-B FCGR3B (SL008609) had a unique association with the percentage of A2(6)BG1 in total plasma N-glycans (GP3). SLAMF7 (SL016928) SLAM family member 7 showed a positive association with the percentage of FA2BG2S(3,6)2 + FA2BG2S(6,6)2 + FA2BG2S(3,3)2 in total plasma N-glycans (GP21) and negative with the percentage of A2G2S(3,6)2 + A2G2S(6,6)2 + A2G2S(3,3)2 in total plasma N-glycans (GP19).

## 3. Discussion

We report here what is to the best of our knowledge the first association study between the blood circulating proteome and its corresponding N-glycome. We observed correlations between N-glycans and proteins that confirm previous observations, and also found numerous novel associations that require future experimental validation.

We are aware of the following caveats and limitations: As a population-based high-throughput experiment, this study only provides a first step towards a better understanding of the interplay of glycans and proteins. Targeted mass-spectrometry methods can for instance be deployed to confirm and refine the protein-glycosylation patterns we report here. We chose not to correct for any confounding factors that are known to influence protein glycosylation, such as age, sex, and diabetes. These factors are creating the diversity that we probe by analysing protein–glycan correlations. Independent studies with larger sample sizes are needed to further investigate their impact on specific protein glycosylation patterns. The fact that the samples were collected from diverse populations increases the variability in the data set and may have lessened the statistical power of the associations; the uncovered correlations may thus be expected to be robust and generalizable across populations. The size of the replication cohort was relatively small and did not allow for the application of a strict Bonferroni level. However, by requiring Bonferroni significance in the discovery study, we believe that we limited the number of false positives to a minimum, an assumption that is confirmed by the very low level of discordant associations that reached nominal significance in the replication study, and by the excellent agreement between QMDiab and TwinsUK for the CRP association, where we had larger sample numbers available.

## 4. Materials and Methods

### 4.1. Study Populations

The Qatar Metabolomics Study on Diabetes (QMDiab) is a cross-sectional diabetes case-control study conducted between February and June 2012 at the Dermatology Department of HMC in Doha, Qatar. QMDiab was approved by the Institutional Review Boards of HMC and Weill Cornell Medicine, Qatar (WCM-Q) (research protocol #11131/11). Written informed consent was obtained from all participants. The study enrolled 369 participants from mainly Arab, South Asian, and Filipino descent who were between 23 and 71 years old [25]. The TwinsUK cohort is a registry of approximately 12,000 adult British twins, mostly female, recruited from the general population through national media campaigns and representative of the UK population in terms of disease-related and lifestyle characteristics [36]. All participants gave written, informed consent, and the Guy’s and St Thomas’ Hospital ethics committee approved the study. A total of 46 female subjects were characterised for both total plasma N-glycans and proteomics data. 

### 4.2. Sample Collection

In the QMDiab study, blood was drawn in the afternoon, after the general operating hours of the morning clinic, into EDTA containers and processed using standardized protocols. Participants were enrolled as they became available, in a random pattern with respect to age, gender, diabetes state, and ethnicity. All procedures were conducted at the same location using identical protocols, instruments, and study personnel. Lab personnel were blinded to all phenotype information. Samples were stored on ice and processed within six hours after sample collection. Blood samples were centrifuged at 2500 *g* for 10 min, aliquoted, and then stored at −80 °C until analysis. For the TwinsUK study, blood samples were taken after at least 6 h of overnight fasting. Serum and plasma EDTA samples were spun at 3000 rpm for 10 min, aliquoted, and stored at −80 °C.

### 4.3. Proteomics Measurements

Proteins were measured using the SOMAscan assay as previously described [6,8]. This technique is based on quantifying protein-specific aptamer binding using a DNA micro-array. Version 3 of the SOMAscan assay convers 1129 distinct proteins. A total of 352 previously unthawed aliquots of 200 μL of EDTA plasma from QMDiab participants was analysed at the WCM-Q proteomics core facility using this assay. The experiments were conducted following Somalogic Inc. protocols, using Somalogic certified instrumentation, and under the direct supervision of experienced Somalogic personnel. Primary data were sent to Somalogic for processing. This includes across-batch calibration and several steps of quality control. No samples or individual probe data were excluded. TwinsUK samples were analysed at the Somalogic laboratory (Boulder, CO, USA). Only proteins available for both cohorts were analysed (N = 1116, Appendix A). 

### 4.4. Total Plasma N-Glycosylation Measurements

Unthawed aliquots of EDTA plasma from QMDiab and TwinsUK were analysed by Genos Ltd. (Zagreb, Croatia) using ultra-performance liquid chromatography (UPLC) glycoprofiling. Glycans were released from total plasma proteins and labelled as described previously [37]. Briefly, 10 μL of plasma sample was denatured with the addition of 20 μL 2% (*w/v*) SDS (Invitrogen, Carlsbad, CA, USA) and N-glycans were released with the addition of 1.2 U of PNGase F (Promega, Madison, WI, USA). The released N-glycans were labelled with 2-aminobenzamide (Sigma-Aldrich, St. Louis, MO, USA). Free label and reducing agent were removed from the samples using hydrophilic interaction liquid chromatography solid-phase extraction. A 0.2 µm 96-well GHP filter-plate (Pall Corporation, USA) was used as stationary phase. Samples were loaded into the wells and after a short incubation washed 5× with cold 96% acetonitrile (ACN). Glycans were eluted with 2 × 90 μL of ultrapure water after 15 min shaking at room temperature, and combined eluates were stored at −20 °C until use. Total plasma N-glycans were then analysed by hydrophilic interaction ultra-performance liquid chromatography (HILIC-UPLC) as described previously [37]. Briefly, fluorescently labelled N-glycans were detected on an Acquity UPLC instrument (Waters, USA) using excitation and emission wavelengths of 250 and 428 nm, respectively. Labelled N-glycans were separated on a Waters BEH Glycan chromatography column, 150 × 2.1 mm i.d., 1.7 μm BEH particles, with 100 mM ammonium formate, pH 4.4, as solvent A and ACN (Fluka, USA) as solvent B. The separation method used a linear gradient of 30–47% solvent A at flow rate of 0.56 mL/min in a 23 min analytical run. Total plasma N-glycans were separated into 39 chromatographic peaks (Figure 4) and then further quantified and annotated into 36 primary and 77 derived glycan traits (Appendix A) [33]. Abbreviations are as follows: all N-glycans have core sugar sequence consisting of two *N*-acetylglucosamines (GlcNAc) and three mannose residues; F indicates a core fucose α1–6 linked to the inner GlcNAc; Ax indicates the number of antennas (GlcNAc) on trimannosyl core; Gx indicates the number of β1–4 linked galactoses on antenna; G1 indicates that the galactose is on the antenna of the α1–6 mannose; Sx indicates the number (x) of sialic acids linked to galactose. Structures in each peak were derived according to Saldova et al. [38].

### 4.5. Statistical Analyses

All statistical analyses were conducted using base libraries in R (version 3.2) [39].

## Figures and Tables

**Figure 1 metabolites-09-00122-f001:**
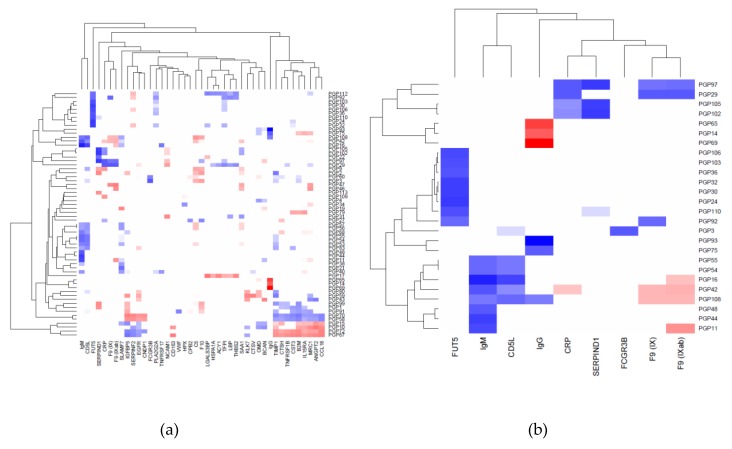
Replicated protein–glycan associations. (**a**) Correlation (r^2^) between 62 glycan (rows) and 43 protein traits (columns); positive associations are in blue, negative in red; darker values indicate stronger associations; (**b**) limited to proteins and glycans that have at least one association with r^2^ > 0.4. A fully annotated and filterable matrix for all associations is available in Excel format as Appendix A.

**Figure 2 metabolites-09-00122-f002:**
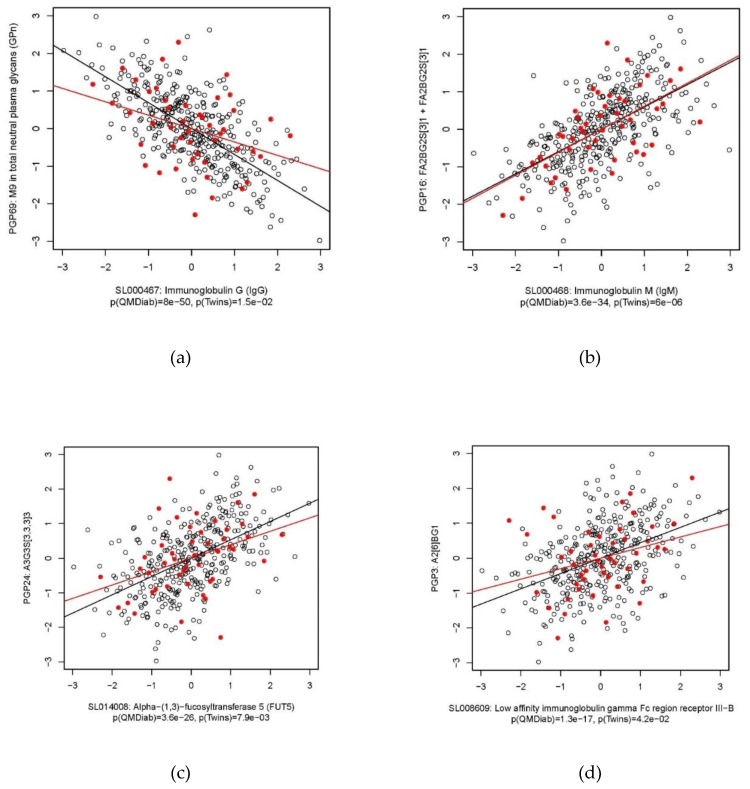
Scatterplots of selected protein–glycan associations. QMDiab (black circles), TwinsUK (red dots); (**a**) IgG with percentage of M9 in total neutral plasma glycans, (**b**) IgM with FA2BG2 glycans, (**c**) FUT5 with A3G3S(3,3,3)3 glycans, and (**d**) FCGR3B with A2(6)BG1 glycans.

**Figure 3 metabolites-09-00122-f003:**
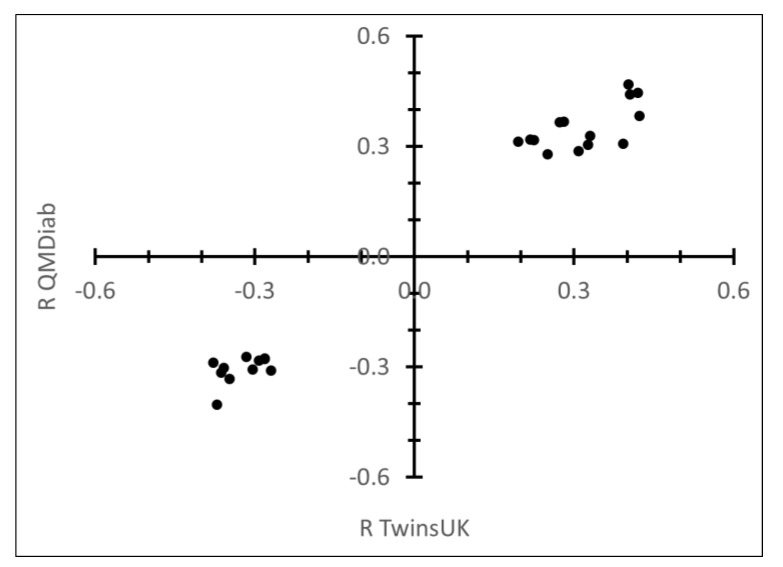
Consistency of the association between glycan traits and C-reactive protein (CRP) levels in TwinsUK and QMDiab. Correlation coefficients for TwinsUK and QMDiab are shown.

**Figure 4 metabolites-09-00122-f004:**
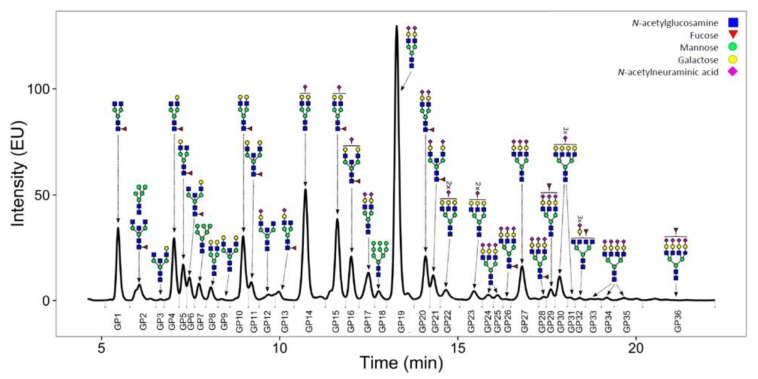
Representative chromatogram of a total plasma N-glycome. Fluorescently labelled plasma N-glycans were separated by HILIC-UPLC into 36 peaks (GP1–GP36). The glycan content in each peak was assigned as determined previously [38]. The amount of glycan species in each peak was expressed as % of total integrated area.

**Table 1 metabolites-09-00122-t001:** Selected protein–glycan associations. Pearson correlations between inverse-normal scaled traits and corresponding *p*-values are listed.

Protein	Glycan Trait	R QMDiab	R TwinsUK	*p*-Value QMDiab	*p*-Value TwinsUK
IgG	PGP69: M9 in total neutral plasma glycans (GPn)	−0.69	−0.36	8.0 × 10^−50^	1.5 × 10^−2^
IgG	PGP93: core fucosylated structures	0.68	0.31	3.7 × 10^−47^	3.7 × 10^−2^
IgG	PGP65: A2G2 in total neutral plasma glycans (GPn)	−0.51	−0.35	3.8 × 10^−24^	1.6 × 10^−2^
IgG	PGP75: fucosylation of digalactosylated structures in total neutral plasma glycans	0.44	0.35	1.3 × 10^−17^	1.9 × 10^−2^
IgG	PGP14: A2G2S(6)1 + A2G2S(3)1	−0.44	−0.30	1.4 × 10^−17^	4.3 × 10^−2^
IgM	PGP16: FA2BG2S(3)1 + FA2BG2S(6)1	0.59	0.61	3.6 × 10^−34^	6.0 × 10^−6^
IgM	PGP44: monosialylation of core-fucosylated digalactosylated structures with bisecting GlcNAc	0.51	0.48	2.5 × 10^−24^	8.0 × 10^−4^
IgM	PGP11: FA2BG2	0.48	0.32	1.3 × 10^−21^	2.9 × 10^−2^
IgM	PGP42: monosialylation of core-fucosylated digalactosylated structures without bisecting GlcNAc	0.46	0.41	4.6 × 10^−19^	4.1 × 10^−3^
IgM	PGP48: ratio of fucosylated monosialylated and disialylated structures (with bisecting GlcNAc)	0.45	0.38	1.7 × 10^−18^	8.3 × 10^−3^
IgM	PGP54: ratio of fucosylated monosialylated structures with and without bisecting GlcNAc	0.40	0.56	1.1 × 10^−14^	4.4 × 10^−5^
IgM	PGP55: the incidence of bisecting GlcNAc in all fucosylated monosialylated structures	0.40	0.56	1.1 × 10^−14^	4.4 × 10^−5^
SERPIND1	PGP97: trisialylated structures	0.53	0.29	8.0 × 10^−27^	4.9 × 10^−2^
SERPIND1	PGP102: trigalactosylated structures	0.53	0.35	8.7 × 10^−26^	1.7 × 10^−2^
SERPIND1	PGP105: triantennary structures	0.52	0.34	7.0 × 10^−25^	2.0 × 10^−2^
FUT5	PGP24: A3G3S(3,3,3)3	0.53	0.39	3.6 × 10^−26^	7.9 × 10^−3^
FUT5	PGP30: A4G4S(3,3,3)3	0.51	0.50	3.8 × 10^−24^	3.8 × 10^−4^
FUT5	PGP32: A4F1G3S(3,3,3)3 + A4F1G3S(3,3,6)3 + A4F1G3S(3,6,6)3	0.51	0.43	4.4 × 10^−24^	3.1 × 10^−3^
FUT5	PGP106: tetraantennary structures	0.48	0.46	1.9 × 10^−21^	1.4 × 10^−3^
FUT5	PGP110: ratio of trisialylated and tetrasialylated tetragalactosylated structures	0.47	0.48	9.4 × 10^−21^	8.3 × 10^−4^
FUT5	PGP103: tetragalactosylated structures	0.47	0.45	2.4 × 10^−20^	1.5 × 10^−3^
FUT5	PGP36: A4F1G4S(3,3,3,6)4	0.45	0.30	6.3 × 10^−19^	4.0 × 10^−2^
FUT5	PGP92: antennary fucosylated structures	0.40	0.37	6.6 × 10^−15^	1.1 × 10^−2^
CD5L	PGP16: FA2BG2S(3)1 + FA2BG2S(6)1	0.51	0.40	6.3 × 10^−24^	5.7 × 10^−3^
CD5L	PGP108: glycan structures with bisecting GlcNAc	0.40	0.41	6.7 × 10^−15^	4.9 × 10^−3^
CRP	PGP97: trisialylated structures	0.45	0.38	3.3 × 10^−18^	8.8 × 10^−3^
CRP	PGP29: A3F1G3S(3,3,3)3 + A3F1G3S(3,3,6)3	0.44	0.39	7.1 × 10^−18^	7.1 × 10^−3^
F9 (IX)	PGP29: A3F1G3S(3,3,3)3 + A3F1G3S(3,3,6)3	0.44	0.38	6.9 × 10^−18^	9.7 × 10^−3^
F9 (IXab)	PGP29: A3F1G3S(3,3,3)3 + A3F1G3S(3,3,6)3	0.44	0.38	7.1 × 10^−18^	9.3 × 10^−3^
F9 (IX)	PGP92: antennary fucosylated structures	0.41	0.31	4.5 × 10^−15^	3.5 × 10^−2^
FCGR3B	PGP3: A2(6)BG1	0.44	0.30	1.3 × 10^−17^	4.2 × 10^−2^

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
