# Peer review of "Fine-Mapping of the Human Blood Plasma N-Glycome onto Its Proteome"

_metabolites, 2019, doi:10.3390/metabo9070122_

Round 1
Reviewer 1 Report
In the paper titled "Fine-mapping of the human blood plasma N-glycome onto its proteome" by Suhre et al, authors analyze the relationship between proteins and N-glycans of human plasma by finding correlation between the levels of expression of particular proteins and N-glycans and analyze for correlation. They argue that the correlation indicates either (1) a physical link, where the protein and the glycan are a part of a single glycoprotein, (2) the protein is an enzyme involved in glycan biosynthesis, (3) the glycan controls the abundance of the protein in the blood or (4) both glycan and the protein are controlled by a confounding factor – such as age and/or disease state, smoking and such – therefore the correlation is based in biological events. Authors use a novel aptamer-based SOMAscan chip in order to measure protein concentrations over a wide concentration range. N-glycan concentrations are measured chromatographically, using HILIC-UPLC after enzymatic digestion. The authors find correlation between 43 proteins and 62 glycans and report in detail on 5 selected proteins (IgG, IgM, heparin cofactor 2, alpha1,3 fucosyltransferase and CRP).
The manuscript is written in a clear manner and the data are concise. Caveats and limitations are discussed.
I suggest accepting this article in present form.
Author Response
We thank you reviewer for their time and evaluation.
Reviewer 2 Report
This study presents the analysis of correlations between the levels of serum proteins and N-glycans in blood samples from two populations of donors or patients obtained from the Dermatology Department of HMC in Doha, Qatar and from the TwinsUK cohort tissue bank. The authors used techniques which were described in their previous publication or references. Overall, the paper reports interesting correlations between abundancy of at least 8 proteins in blood samples and different types of proceeded glycans. The manuscript is well-written however several points are required for clarification or editing:
1. Description of specific glycans is not consistent through the text and mixed terms/abbreviations are used. For example, compare the abbreviations in Figure 5 (GP etc) and Table 1/Figure 1 (PGP etc). Obviously the authors refer to the technical information of their lab records and experiments, which has no meaning for the scientific text as per se.
2. Reference to Table 1 is missing in the text.
2. Figure 2: X-axes titles are weird and have to be revised.
3. The authors refer to supplementary Tables, however they do not present any data in these tables, just annotations of what they used for data analysis. Supplements section can be also used to show the structure of glycans specified in the Table 1.
4. References: #16 has extra initials, #22 has unknown stars.
Author Response
We thank you reviewer for their time and evaluation.
Response to point 1.: We apologize for this confusion. The uploaded Supplement erroneously only consisted of the table of content in PDF format, not the Supplemental Tables themselves. Therefor the relevant information regarding the glycan annotations was not available. There are 113 glycan traits, referred to by id as PGP1 to PGP113. The first 36 of these traits correspond to the individual glycan peaks GP1-GP36. Their exact nomenclature is given in Supplementary Table 2. Throughout the manuscript we use glycan ids (PGPn) to refer to specific glycan traits, and when relevant, we also provide their annotation.
Response to point 2: Please excuse this omission. We added a call-out to Table 1 at the relevant position at the end of the first result paragraph.
Response to point 2b: Some of the x-titles were too long and had been truncated. We revised the titles to fit under the graph.
Response to point 3: see response to point 1 – we now provide the full Supplement with all information, including the standard nomenclature that describes the structure of the glycans.
Response to point 4: We corrected the references